# Loss of endocytosis-associated RabGEF1 causes aberrant morphogenesis and altered autophagy in photoreceptors leading to retinal degeneration

**Passley Hargrove-Grimes**[1,2], **Anupam K. Mondal**[1⊙], **Jessica Gumerson**[1⊙], **Jacob Nellissery**[1], **Angel M. Aponte**[3], **Linn Gieser**[1], **Haohua Qian**[4], **Robert N. Fariss**[5], **Juan S. Bonifacino**[6], **Tiansen Li**[1], **Anand Swaroop**[1]*

1 Neurobiology, Neurodegeneration & Repair Laboratory, National Eye Institute, National Institutes of Health, Bethesda, Maryland, United States of America, 2 Institute of Biomedical Sciences, George Washington University, Washington, District of Columbia, United States of America, 3 Proteomics Core, National Heart Lung and Blood Institute, National Institutes of Health, Bethesda, Maryland, United States of America, 4 Visual Function Core, National Eye Institute, National Institutes of Health, Bethesda, Maryland, United States of America, 5 Biological Imaging Core, National Eye Institute, National Institutes of Health, Bethesda, Maryland, United States of America, 6 Neurosciences and Cellular and Structural Biology Division, Eunice Kennedy Shriver National Institute for Child Health and Development, National Institutes of Health, Bethesda, Maryland, United States of America

⊙ These authors contributed equally to this work.
* swaroopa@nei.nih.gov

**Data Availability Statement:** Raw and processed RNA seq datasets from this study have been

## Abstract

Rab-GTPases and associated effectors mediate cargo transport through the endomembrane system of eukaryotic cells, regulating key processes such as membrane turnover, signal transduction, protein recycling and degradation. Using developmental transcriptome data, we identified *Rabgef1* (encoding the protein RabGEF1 or Rabex-5) as the only gene associated with Rab GTPases that exhibited strong concordance with retinal photoreceptor differentiation. Loss of *Rabgef1* in mice (*Rabgef1⁻/⁻*) resulted in defects specifically of photoreceptor morphology and almost complete loss of both rod and cone function as early as eye opening; however, aberrant outer segment formation could only partly account for visual function deficits. RabGEF1 protein in retinal photoreceptors interacts with Rabaptin-5, and RabGEF1 absence leads to reduction of early endosomes consistent with studies in other mammalian cells and tissues. Electron microscopy analyses reveal abnormal accumulation of macromolecular aggregates in autophagosome-like vacuoles and enhanced immunostaining for LC3A/B and p62 in *Rabgef1⁻/⁻* photoreceptors, consistent with compromised autophagy. Transcriptome analysis of the developing *Rabgef1⁻/⁻* retina reveals altered expression of 2469 genes related to multiple pathways including phototransduction, mitochondria, oxidative stress and endocytosis, suggesting an early trajectory of photoreceptor cell death. Our results implicate an essential role of the RabGEF1-modulated endocytic and autophagic pathways in photoreceptor differentiation and homeostasis. We propose that RabGEF1 and associated components are potential candidates for syndromic traits that include a retinopathy phenotype.

submitted to Gene Expression Omnibus (GEO) under the accession GSE138672. (https://www.ncbi.nlm.nih.gov/geo/query/acc.cgi?acc=GSE138672).

**Funding:** These studies were supported by Intramural Research program of the National Eye Institute (ZIAEY000450 and ZIAEY000546 to AS). The funders had no role in study design, data collection and analysis, decision to publish, or preparation of the manuscript.

**Competing interests:** The authors have declared that no competing interests exist.

## Author summary

Endocytosis and autophagy are evolutionarily conserved processes that are essential for maintenance of cellular homeostasis. RabGEF1 is a major regulator of the Rab5-GTPase, which participates in key steps during endocytosis and autophagy. We demonstrate that loss of RabGEF1 in mice causes specific developmental defects during photoreceptor outer segment formation, leading to visual dysfunction as early as eye opening followed by retinal degeneration. *Rabgef1⁻/⁻* retina shows a clear reduction in early endosomes as well as accumulation of autophagic vacuoles in developing photoreceptors. Together with transcriptome analysis, our studies suggest a trajectory of cellular events including altered autophagy that precede photoreceptor cell death in the absence of RabGEF1 and establish a critical role of endocytosis and autophagy in retinal development and proteostasis.

## Introduction

Despite extensive phenotypic and genetic heterogeneity, functional defects and/or death of photoreceptors are the primary cause of vision impairment in retinal and macular degenerative diseases [1–3] (https://sph.uth.edu/retnet/). The retinal photoreceptors, rods and cones, are highly metabolically active sensory neurons that are responsible for dim and daylight vision, respectively. Rod and cone photoreceptors initiate vision by photon capture in morphologically distinct modified primary cilia (called outer segments) containing stacks of membranous discs with embedded opsin visual pigment(s) and transmit visual information to inner retina neurons through ribbon synapses [4,5]. Differentiation of these structurally and functionally unique cells in the developing mammalian retina spans a long period (3–4 weeks in mice and several months in humans) and is dictated by a select subset of transcriptional regulatory proteins [6,7]. Circadian-controlled shedding and subsequent renewal of outer segment discs are critical for functional maintenance of photoreceptors and involve extensive membrane turnover, polarized protein transport and stringent control of all facets of intracellular trafficking [8]. Not surprisingly, genetic defects affecting outer segment biogenesis and ciliary transport processes constitute almost one-third of all inherited retinal diseases [9–12].

   Endocytosis is an essential evolutionarily conserved process for internalization of extracellular molecules, signal transduction and turnover of plasma membrane components, which are critical for maintaining cellular functions and homeostasis [13–16]. Distinct steps of the endocytic pathway occur in specific subcellular compartments and involve a number of small GTPases and associated proteins; of these, Rab (**Ra**s-Related Proteins in the **B**rain) proteins and their effectors are implicated in almost all aspects of this intricate process [17]. Like other small monomeric GTPases of the Ras superfamily, Rabs are ubiquitously expressed binary molecular switches, cycling between an inactive GDP-bound state in the cytosol and an active GTP-bound state on vesicle membranes [17,18]. The cycle of Rab activation involves the recruitment of a wide array of effector molecules to control vesicular trafficking within the endocytic pathway including but not limited to vesicle budding, motility, tethering and fusion [19,20]. Dysfunction of Rabs or their effectors results in diverse clinical phenotypes affecting nearly all tissues and organ systems [21,22]. Mutations in two of the Rab family proteins, Rab27 and Rab28, have been associated with retinal dystrophies [23,24]. Notably, Rab28 is required for turnover of endocytosed proteins and for lysosomal delivery of protein cargo in *Trypanosoma brucei* [25]. However, the relationship of Rabs and their effectors to endocytosis has not been explored in the retina.

Studies in *Xenopus* retina provided the first direct evidence of endocytosis in vertebrate rod photoreceptors to presumably recover and recycle molecular components from the inner segment plasma membrane [26]. In later studies, synaptic vesicle endocytosis was also demonstrated in Salamander cone photoreceptors as a likely mechanism to maintain neurotransmitter release at a high physiological rate [27]. Additionally, maintenance of rhabdomeres (outer segment-like structures) in *Drosophila* photoreceptors reportedly requires endocytosis [28]. However, we currently have a poor understanding of the endocytic pathway and its relevance in the mammalian retina, especially, in photoreceptors.

Advances in next generation sequencing technology have permitted transcriptome profiling of different tissues and cell types even at a single cell level. Taking advantage of the published transcriptome datasets (https://neicommons.nei.nih.gov/#/analysis), we identified the endocytic regulator *Rabgef1* as the only Rab-associated gene showing increasing expression during retinal development. Herein we show that the *Rabgef1*[-/-] retina exhibits little visual function at eye opening in mice, with relatively fast photoreceptor degeneration due to compromised biogenesis of outer segment discs and accumulation of autophagosome-like vesicles within the inner segment region. Based on immunostaining and retinal transcriptome analysis, we propose a likely cellular sequence of events including decreased autophagic flux that lead to photoreceptor cell death in the absence of RabGEF1. Our studies provide direct support for the centrality of RabGEF1, and consequently its target Rabs, in endocytic and/or autophagic events during functional maturation and homeostasis of retinal photoreceptors. We suggest that RabGEF1 and related components should be considered as candidates for human retinopathies and/or syndromes that include a photoreceptor degeneration phenotype.

## Results

### *Rabgef1* expression during retinal differentiation

We examined the expression of all Rabs and related effector proteins in transcriptome datasets of developing mouse retina [29,30] from embryonic day 11 (E11) to postnatal day 28 (P28) and identified *Rabgef1* as the only Rab-associated gene that showed increasing expression from P6 onwards (S1 Table). RabGEF1 functions during endocytosis as a GEF of Rab5 through formation of an obligatory and tight physical complex with Rabaptin-5 [31–33] (Fig 1A). When activated by the RabGEF1-Rabaptin5 complex, Rab5-GTP interacts with a variety of effector proteins including tethering factors such as Early Endosomal Antigen 1 (EEA1) and SNARE proteins such as Syntaxin13, leading to heterotypic fusion of clathrin-coated vesicles with early endosomes or homotypic fusion of early endosomes with each other [14,34] (Fig 1A). Activation of Rab5 by RabGEF1 promotes initial stages of the endocytic pathway, propelling cargoes along degradation or recycling routes [20]. Expression of *Rabgef1* transcripts in the developing mouse retina paralleled the trajectory of rod and cone photoreceptor differentiation [35,36] (Fig 1B). *In situ* hybridization experiments revealed *Rabgef1* mRNA in all retinal cell types, with strong expression noted within the outer nuclear layer (ONL) of photoreceptors (Fig 1C). Immunoblot analysis of mouse retinal extracts demonstrated the peak of RabGEF1 protein expression at P14 (at the time of eye opening), when photoreceptor outer segment formation and synaptogenesis are proceeding rapidly (Fig 1D and S1A Fig).

To elucidate the function of RabGEF1 in the mammalian retina, we examined *Rabgef1*[-/-] (knockout, KO) mice [37] through postnatal development and later ages. As predicted, RabGEF1 protein was undetectable in the KO retina by immunoblotting (Fig 1D) and immunohistochemistry (Fig 1E). In contrast, RabGEF1 expression was readily detectable by immunoblotting of control retinas (*Rabgef1*[+/+], *Rabgef1*[+/-]) (Fig 1D). RabGEF1 protein was also evident in photoreceptor inner segments and synaptic terminals in the outer plexiform layer

(OPL) by immunostaining of control retinas (Fig 1E), though non-specific background staining was also observed in both control and KO samples.

## Loss of RabGEF1 results in photoreceptor degeneration

Histology was performed for control and *Rabgef1*⁻/⁻ mouse retina from developing (P10) to mature stages (P45). At early stages, retinal development appeared normal in *Rabgef1*⁻/⁻ mice

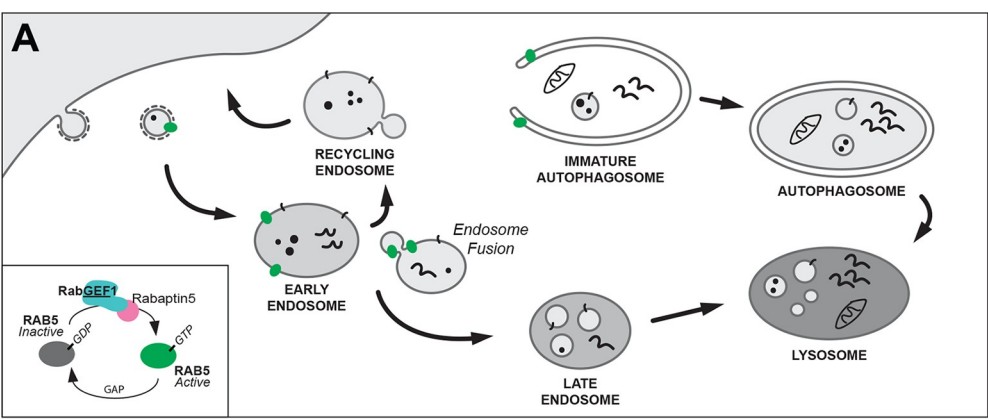

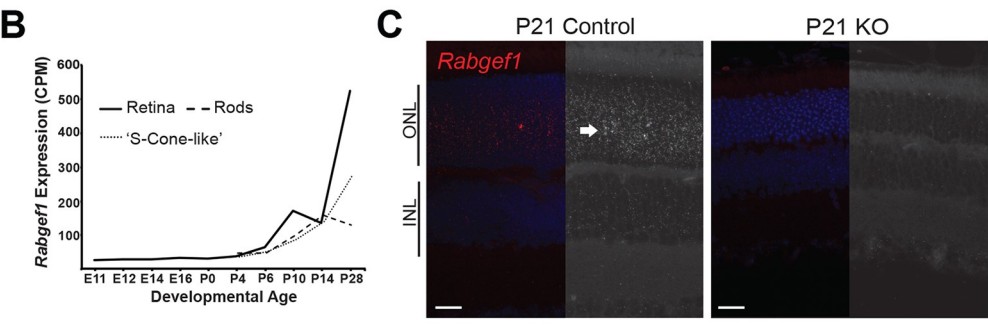

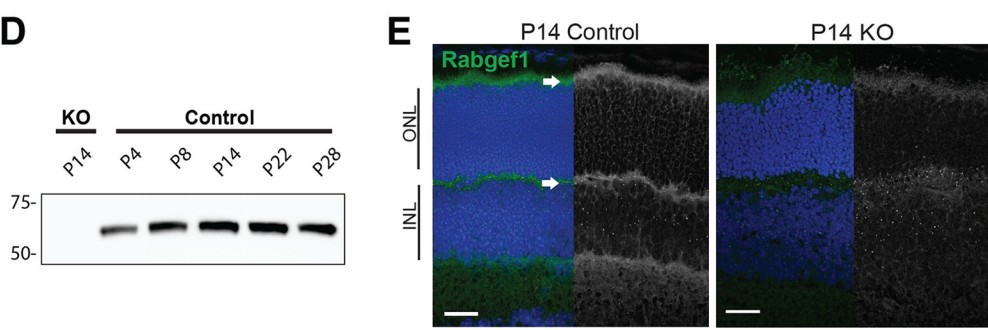

**Fig 1. Spatiotemporal expression of *Rabgef1* in the mouse retina.** (**A**) A schematic showing conserved functions of Rab5/RabGEF1 in endocytosis and autophagosome closure. (**B**) *Rabgef1* expression in developing whole retina or flow-sorted rod and S-cone-like photoreceptors (identified from RNA-seq data in [30,36]). *Nrl*-GFP mice enabled purification of rod photoreceptors (Rods), whereas *Nrl*-GFP mice crossed with the cone-only *Nrl*⁻/⁻ mice allowed sorting of S-cone-like photoreceptors [63]. (**C**) *In situ* hybridization profile of postnatal (P)21 control and *Rabgef1*⁻/⁻ (KO) retina. Red/white punctate dots represent single *Rabgef1* mRNA molecules. Arrow indicates high *Rabgef1* transcripts in photoreceptor layer. Scale bar = 20 μm. ONL, outer nuclear layer; INL, inner nuclear layer. (**D**) Immunoblot analysis of P4 –P28 control and *Rabgef1*⁻/⁻ P14 retinal lysates probed with anti-RabGEF1 antibody. The total protein loading control is included in S1 Fig. (**E**) Immunohistochemistry of P14 control and *Rabgef1*⁻/⁻ retinal sections using anti-RabGEF1 antibody. Arrows indicate high RabGEF1 expression in photoreceptor inner segments and synaptic terminals. Only non-specific background staining (punctate dots), also observed in control, is detected in *Rabgef1*⁻/⁻ retinal sections. DAPI was used for visualizing nuclei. Scale bar = 20 μm.

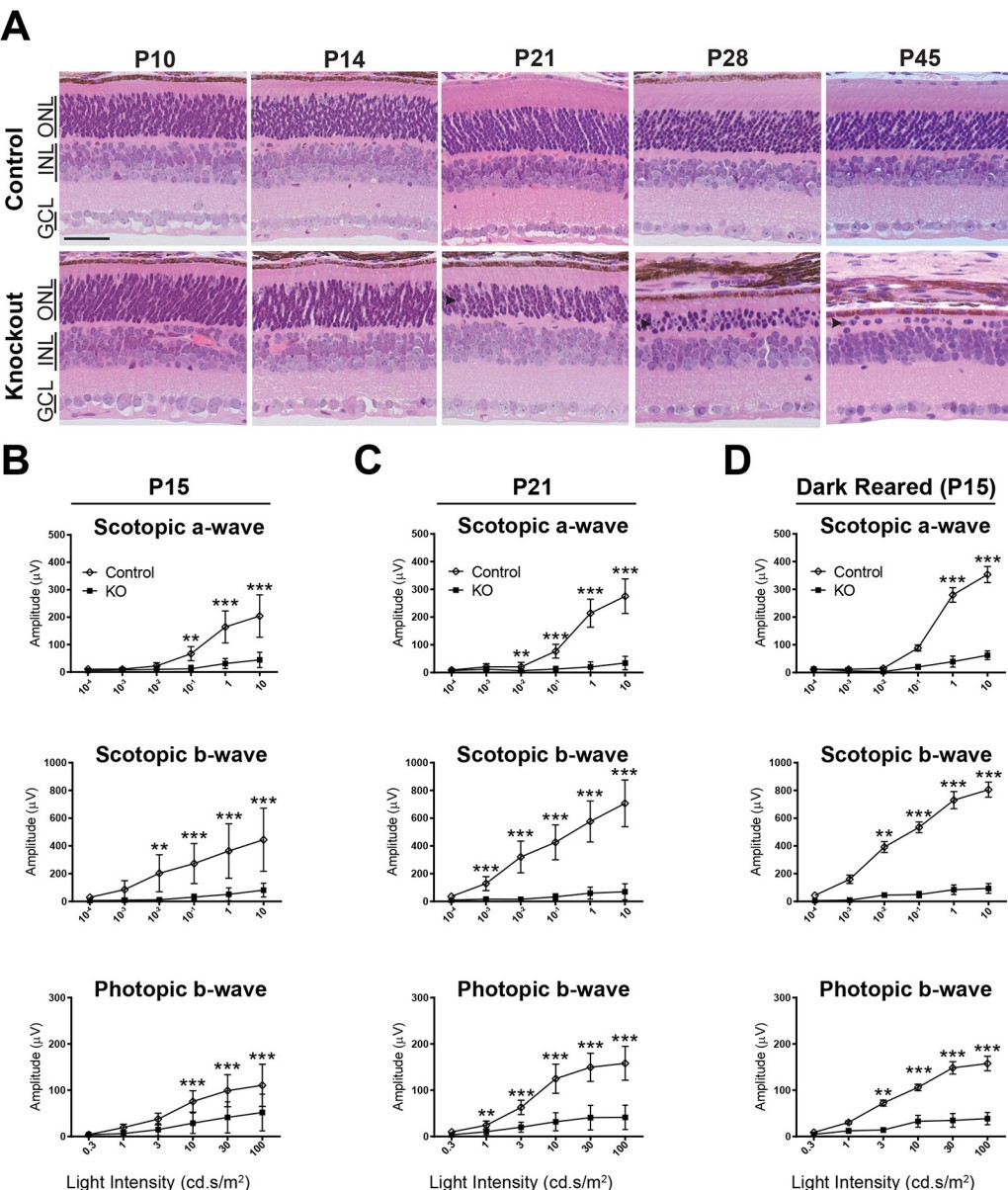

**Fig 2. Rapid photoreceptor degeneration and loss of visual function in *Rabgef1*$^{-/-}$ mice. (A)** H&E staining of developing control and *Rabgef1*$^{-/-}$ retina, showing near complete ablation of photoreceptors (ONL) by postnatal day 45. Scale bar = 50 μm. ONL, outer nuclear layer; INL, inner nuclear layer; GCL, ganglion cell layer. **(B, C)** ERG stimulus intensity-amplitude functions in the control and KO mice reared in a 12h light/12h dark cycle and measured at postnatal day 15 and 21 (P15, P21), respectively. **(D)** ERG stimulus intensity-amplitude functions in control and *Rabgef1*$^{-/-}$ mice born and raised in the dark. Dark-reared *Rabgef1*$^{-/-}$ mice have similar ERG stimulus intensity-amplitude functions as animals reared in cyclic light. Asterisks indicate p-value < 0.05 (*), <0.01 (**) and <0.001 (***) as determined by a t-test using Prizm software. In panels B and C, n = 7–10 per genotype and in D, 2–3 per genotype. Error bars indicate SD.

and was indistinguishable from littermate controls (Fig 2A and S1B Fig). However, the retina of P21 KO mice showed signs of thinning, and specific cell loss was evident within the ONL. Rapid loss of photoreceptors continued with age, and only 1–2 rows of cells remained in the ONL by P45 (Fig 2A and S1 Fig). While rod outer segments were noticeably shorter by P21 as indicated by rhodopsin immunostaining (S2 Fig). Cones, bipolar, amacrine, horizontal and ganglion cells appeared to be largely unaffected in the KO retina at P21 based on immunostaining of cell-type

specific markers (S2 Fig). Notably, S-opsin staining was dramatically reduced at P45 (S2 Fig, right panel). As predicted, immunostaining of ribeye (a marker of photoreceptor synaptic ribbons) was progressively reduced in the OPL, and GFAP reactivity (a marker of activated Müller glia) was elevated in *Rabgef1*-/- retina due to photoreceptor degeneration (S2 Fig).

We performed electroretinogram (ERG) recordings to test for visual function. *Rabgef1*-/- retina showed diminished responses to light stimuli compared to littermate controls at different ages (Fig 2B and S3 Fig). The dark-adapted (scotopic) a-waves reflecting rod photoreceptor function, and the corresponding b-waves originating from inner retina neurons, were dramatically reduced as early as P15 in the mutant retina (Fig 2B), demonstrating functional deficits in rod phototransduction and in synaptic transmission to inner retinal neurons despite apparently normal morphology (see Fig 2A). Compared to littermate controls, *Rabgef1*-/- retina also showed significantly reduced light-adapted (photopic) b-waves, indicating a compromised cone-mediated visual response. By P21, rod responses (scotopic) were barely detectable in KO retina, and cone responses (photopic) were greatly reduced (Fig 2C and S3 Fig). Notably, at least half of the rods remain at this age in the KO retina as indicated by ONL thickness (Fig 2A), suggesting a functional defect in rods beyond the underlying cell loss. In a number of retinal degeneration models, cell loss is accelerated by light exposure. To determine whether the decline in visual function in *Rabgef1*-/- mice was similarly exacerbated by light, ERGs were recorded from animals reared in the dark from birth. No significant effect of dark rearing was detected on rod or cone photoreceptor function in the KO retina (Fig 2D), and ERG traces were comparable to animals reared under standard cyclic lighting (S3 Fig). Thus, visual function decline in the *Rabgef1*-/- retina is independent of light exposure.

## Outer segment biogenesis is compromised in the *Rabgef1*-/- retina

Immunohistochemical analysis of P14 *Rabgef1*-KO retina showed correct localization, yet reduced staining, of key outer segment proteins including rhodopsin, cyclic nucleotide gated channel α and β subunits (Cnga and Cngb) and cyclic GMP phosphodiesterase β (Pde6b) (Fig 3A), suggesting that photoreceptor dysfunction at this early stage (see Fig 2B) cannot be solely attributed to mis-trafficking of phototransduction proteins. Immunoblotting of retinal proteins corroborated the immunohistochemical data and revealed a decrease in many outer segment proteins in *Rabgef1*-KO samples compared to littermate controls (Fig 3B).

To determine why the *Rabgef1*-KO retina exhibits a dramatically reduced visual response at early developmental timepoints despite correct localization of phototransduction proteins, we performed ultrastructural analysis by transmission electron microscopy (TEM). Around the time of eye opening (P14), we noted fewer, stunted and disorganized outer segments in the *Rabgef1*-/- retina compared to controls (Fig 4A). Furthermore, at higher magnification, large electron-dense and amorphous aggregates contained within vacuoles with the appearance of autophagic organelles (black arrowheads) were consistently detected in the inner segments of many photoreceptors in P14 KO retinas (Fig 4B). Additional TEM analysis of *Rabgef1*-/- retinas revealed the presence of these aggregates in a few photoreceptors as early as P10 (S4 Fig), with progressive and rather dramatic increase in both rods and cones by retinal maturity around P22 (Fig 4C and S4 Fig). Overall, these results indicate that the loss of *Rabgef1* specifically leads to early cellular defects in outer segment biogenesis, with concomitant accumulation of amorphous aggregates within the inner segments before photoreceptor degeneration is evident by histology.

## Loss of RabGEF1 results in reduction of early endosomes

To investigate whether RabGEF1 also functions in the endocytic pathway in the mammalian retina, we performed co-immunoprecipitation studies to identify its interacting proteins using

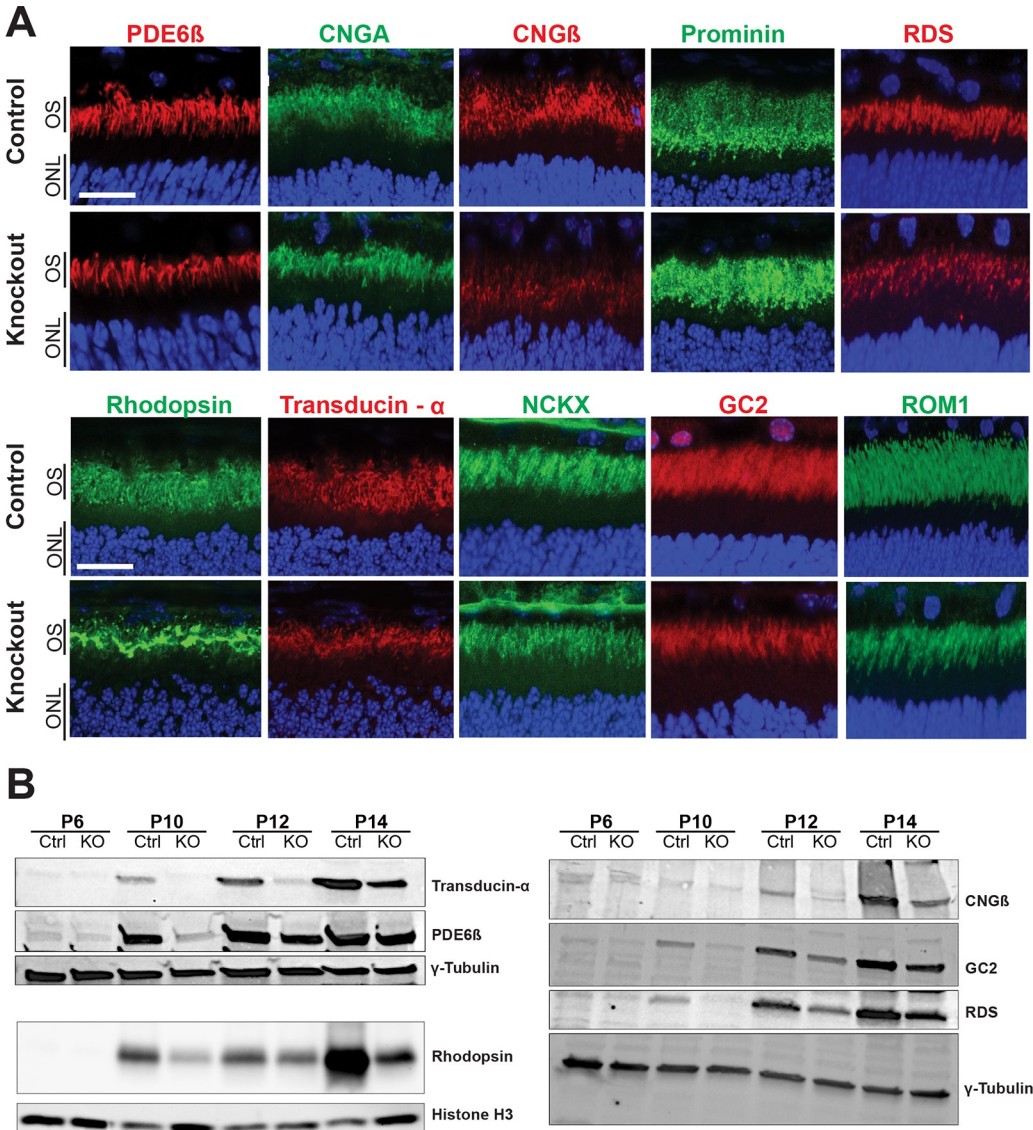

**Fig 3. Reduced immunostaining of outer segment proteins in the *Rabgef1*[-/-] retina. (A)** Immunohistochemistry of P15 retinas using antibodies against proteins of the phototransduction cascade and proteins required for outer segment integrity. Retinal sections were counterstained with DAPI. ONL, outer nuclear layer; OS, outer segments. Scale bar = 20 μm. **(B)** Immunoblots of proteins from retinal lysates through development, from P6—P14. *Rabgef1*[-/-] retinal lysate protein levels appear reduced at all developmental stages compared to littermate controls. γ-tubulin and Histone H3 are used as loading controls. 40 μg of protein loaded per lane.

P14 mouse retina extracts. As predicted, RabGEF1 was immunoprecipitated from control retinal extracts with anti-RabGEF1 antibody but not with control IgG (Fig 5A). Re-probing of the immunoblot revealed co-immunoprecipitation of Rabaptin-5 with RabGEF1, consistent with their coordinated function in endocytosis. We then performed mass spectrometry analysis of the anti-RabGEF1 and IgG-immunoprecipitated proteins from control and KO retinal lysates. Our analysis confirmed an interaction between RabGEF1 and Rabaptin-5 (also known as Rabep1) in a 2:1 stoichiometry (Fig 5B), similar to previously published data [32,33]. We also observed an interaction of RabGEF1 with the Rabaptin-5 paralog Rabep2. Neither Rabaptin-5 nor Rabep2 were detected by co-immunoprecipitation of control retina extract with rabbit

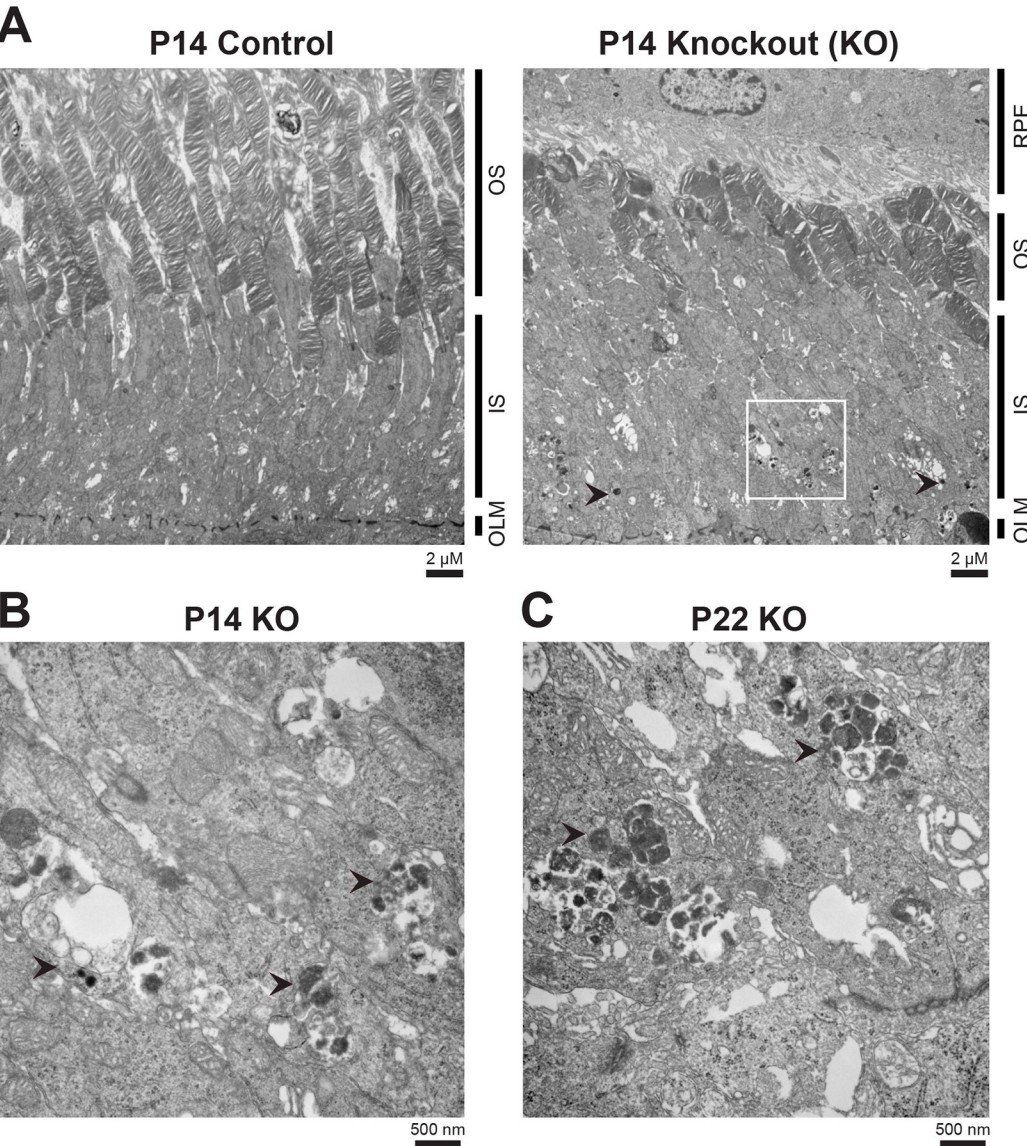

**Fig 4. Loss of *Rabgef1* causes accumulation of amorphous, electron-dense structures in the photoreceptors. (A)** Transmission electron micrographs of control and *Rabgef1⁻/⁻* retinas at around eye opening (P14). White box highlights accumulation of amorphous, electron-dense structures in *Rabgef1⁻/⁻* photoreceptor inner segment regions. **(B)** Magnification of white boxed area in panel A and an image at the same magnification but in P22 *Rabgef1⁻/⁻* retina **(C)**, demonstrating the accumulation of electron-dense deposits in photoreceptor cytoplasm (black arrowheads).

IgG or from *Rabgef1⁻/⁻* retinal lysate with anti-RabGEF1 antibody (Fig 5B). Together, these data suggest that RabGEF1 exerts a conserved endocytic function in the mammalian retina as shown in other somatic cell types.

RabGEF1 is known to be an activator of the Rab5 subfamily. Immunoblot analysis did not show any change in expression of Rab5 in the KO retina (S5A Fig). To evaluate whether Rab5 is functional, we examined control and KO retina by immunostaining for early endosomal antigen 1 (EEA1), a direct effector of activated Rab5-GTP associated with endocytic vesicles during the process of fusion [38]. Interestingly, EEA1+ puncta were present throughout photoreceptor inner segments (white arrows) and in OPL (Fig 5C and S5B Fig) of developing

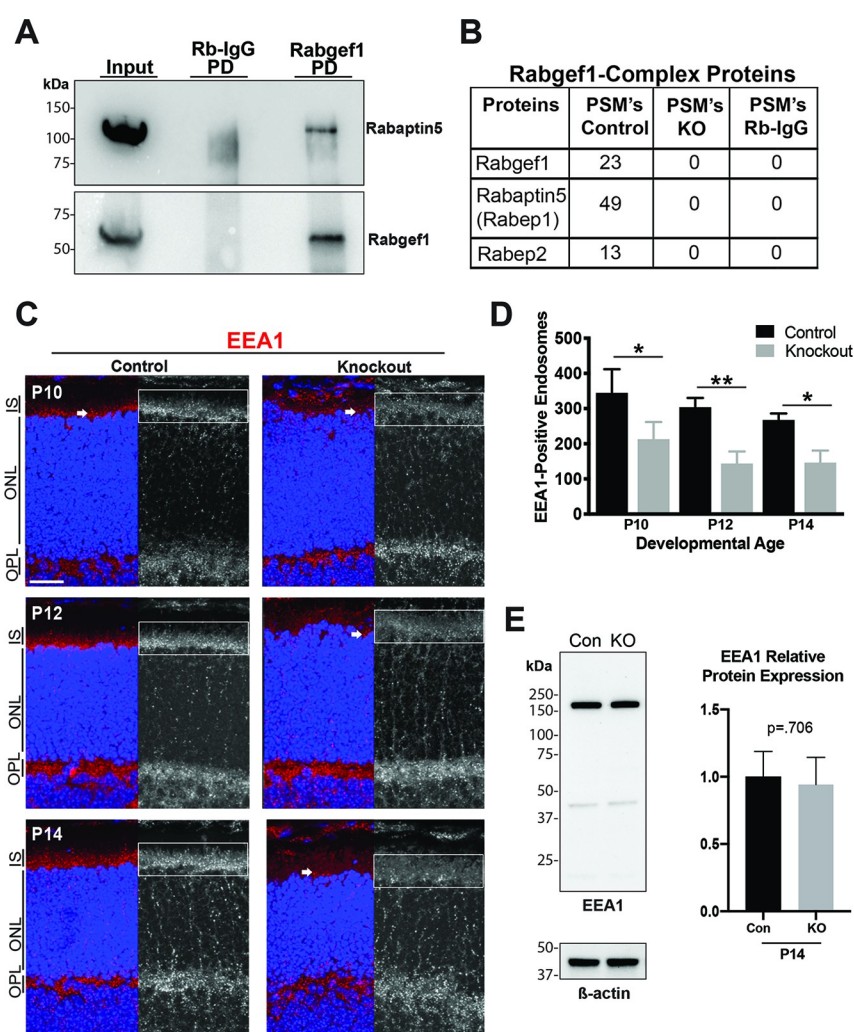

**Fig 5. Conserved endocytic function of RabGEF1 in the retina. (A)** Immunoblot analyses of Rabbit-IgG and anti-RabGEF1 pulldown (PD) protein complexes (from 40 μg retinal lysate) with antibodies against RabGEF1 and Rabaptin5. Input was 3% of the protein from the control P21 mouse retinal lysate. Rabbit-IgG served as a negative control. **(B)** Mass spectrometry analysis of RabGEF1 co-immunoprecipitated proteins from P21 control and *Rabgef1⁻/⁻* retinas. RabGEF1 binds to Rabaptin-5 in a 1:2 ratio respectively (n = 2). **(C)** Immunohistochemistry with anti-EEA1 antibody on control and *Rabgef1⁻/⁻* retinal slices. Retinal sections were counterstained with DAPI. IS, inner segment; ONL, outer nuclear layer; OPL, outer plexiform layer. Scale bar = 20 μm. Boxed areas indicate regions of retina that were quantified, also shown at larger scale in S5B Fig. **(D)** Quantitative measurements of EEA1-positive endosomes at P10, 12 and 14 control and *Rabgef1⁻/⁻* individual z-stack images of the inner segment regions. **(E)** Immunoblot of control and *Rabgef1⁻/⁻* P14 retinal lysates probed with anti-EEA1. Anti-beta-actin was used as a loading control. There is no significant difference between EEA1 protein amounts in control and *Rabgef1⁻/⁻* retinal lysates Asterisks indicate p-value < 0.05 (*) and < 0.01 (**) as determined by t-test using Prizm software. In panels D and E, n = 3 biological replicates. Error bars indicate SD.

control and *Rabgef1⁻/⁻* retina (P10, P12, P14). However, much less EEA1 immunoreactivity (representing reduced membrane association or fewer endocytic vesicles) was detected in the photoreceptor inner segments of KO retina compared to controls. Quantification revealed a significant reduction in the number of EEA1+ vesicles in *Rabgef1⁻/⁻* photoreceptors compared to littermate controls (Fig 5D and S5B Fig). This reduction in EEA1-positive endosomes is likely not due to reduced expression of the EEA1 protein, as immmunoblotting detected similar levels in control and KO retina extracts (Fig 5E). These results suggest a significant loss of

EEA1+ early endocytic vesicles, probably causing perturbation of endocytic fusion in the *Rabgef1*<sup></sup>*-/-* photoreceptors.

## Autophagy is altered in *Rabgef1*-KO photoreceptors

The endocytic machinery in general [16] and RabGEF1 and Rab5 in particular [39,40] play important roles in modulating autophagy. A reduction in early endosomes and existence of vacuoles with electron-dense aggregates in KO photoreceptors suggested autophagy alterations in the absence of RabGEF1. We therefore performed immunohistochemical staining of P10 to P21 control and KO retinas using an antibody to the autophagy marker, microtubule light-chain 3, isoforms A and B (LC3A/B). We observed the presence of LC3A/B-positive autophagosomes specifically in the photoreceptor layer of the KO retina (white arrows) as early as P12, with progressive increase and wider distribution throughout the cell at later stages (Fig 6A). Lower magnification images of LC3A/B staining revealed that autophagosomes in KO retina were restricted to a subset of photoreceptors (ONL and IS layers) and were undetectable in other retinal cell types (Fig 6B). Immunoblot analysis of retinal lysates also indicated an increase in the lipidated, membrane-bound LC3A/B-II form in the *Rabgef1*<sup>-/-</sup> retina (Fig 6C). LC3A/B+ autophagosome structures were not observed in control retina at any developmental stage examined. Moreover, accumulation of LC3A/B+ autophagosomes and/or electron dense structures were not detected in other mouse models of retinal degeneration including the Rhodopsin T17M, *Rd1*, and *Rds* mutants (S5C and S5D Fig), suggesting a specific autophagic alteration in *Rabgef1*-KO retina unrelated to other mechanisms of photoreceptor cell death.

Immunostaining for the autophagy cargo adapter marker p62 (also known as SQSTM1) demonstrated a similar pattern with positive labeling restricted to the photoreceptor layer of KO retina (Fig 6D). High-resolution imaging of p62 staining using the Zeiss LSM880 Airyscan microscope (Fig 6D, right panels) showed a localization pattern reminiscent of the electron-dense structures observed in TEM images (see Fig 4). To determine whether these structures were indeed autophagic vacuoles, post-embedding immunoelectron microscopy was performed with anti-p62 antibody. Indeed, we observed p62 immunogold labeling associated with electron-dense amorphous deposits in the inner segments of KO photoreceptors (Fig 6E). p62 immunostaining was not detected in photoreceptors of control retina by either confocal or electron microscopy. Immunoblot analysis of control and KO retinal protein lysates revealed no appreciable difference in p62 (Fig 6F), probably because p62 accumulation was limited to a subset of photoreceptor cells.

We then isolated vesicular fractions from control and *Rabgef1*<sup>-/-</sup> retinas. Mass spectrometry analysis of these fractions identified 1176 proteins in the control and 1085 in the KO, with significant differences between the two fractions (S6A Fig and S2 Table). Quantitative comparison and Gene Set Enrichment Analysis revealed the absence of or a significant decrease in many proteins associated with phototransduction (such as Gnat1/2, Pde6a, Grk1 and Sag), mitochondrial homeostasis (including Atp5l, Atp5h, Dlat, Cisd1), and synaptic transmission (e.g., Syn1, Syn2, SV2a, SV2b, Snap25) in the KO vesicle fraction (S6B and S6C Fig and S2 Table).

## Transcriptome profiling identifies early changes in developing *Rabgef1*<sup>-/-</sup> retina

To gain insights into early cellular changes resulting from the loss of RabGEF1, we performed comparative analysis of *Rabgef1*-KO versus control retinal transcriptomes at three key developmental ages: P6, P10 (before degeneration is evident), and P14 (when photoreceptors show ultrastructural and visual function defects but without cell loss). Differential expression (DE)

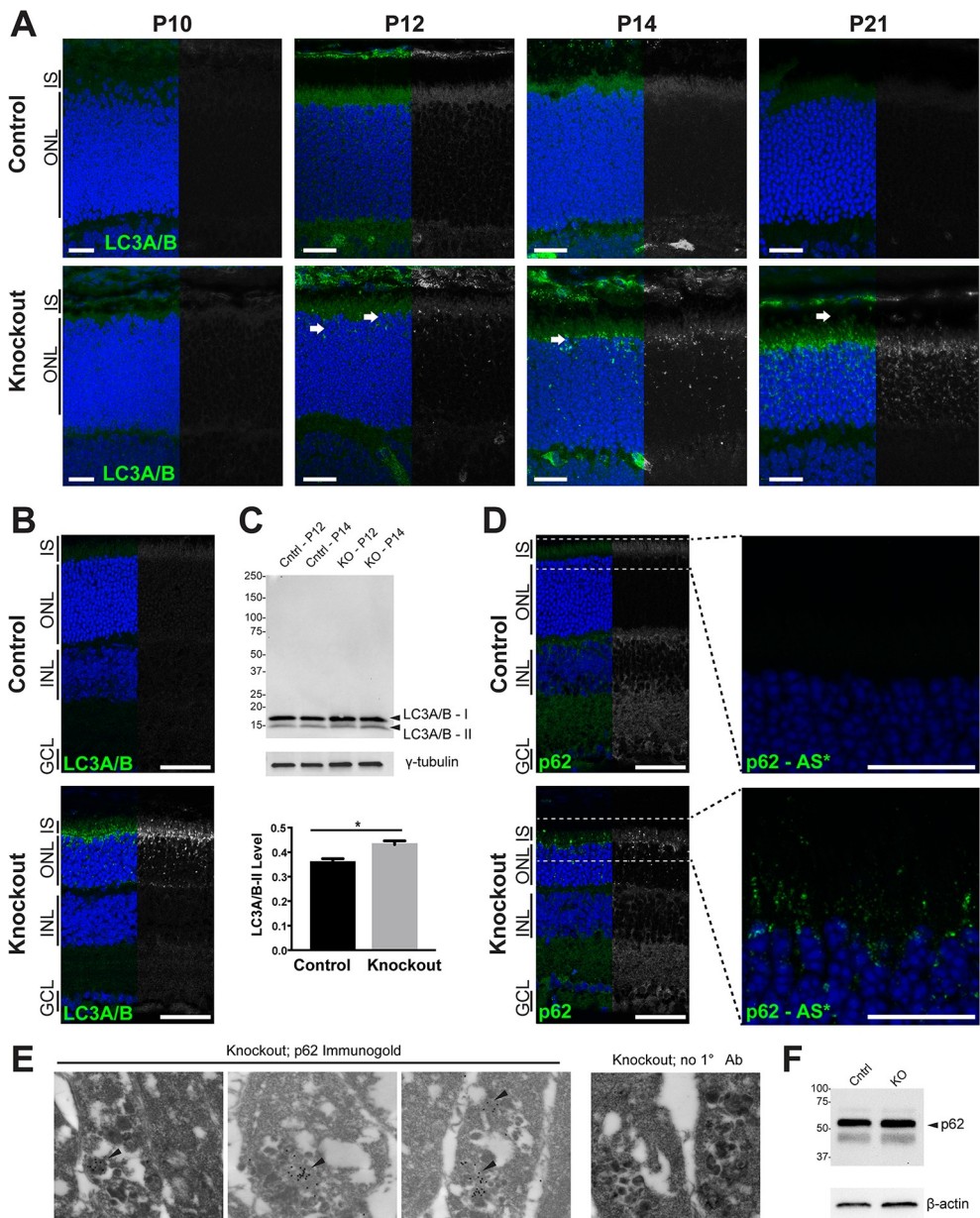

**Fig 6. Loss of *Rabgef1* leads to autophagy defects in the retinal photoreceptors. (A)** Immunohistochemistry of control and *Rabgef1*⁻/⁻ retinas using anti-LC3A/B antibody. Retinal sections were counterstained with DAPI. Scale bar = 20 μm. ONL, outer nuclear layer; IS, inner segment. **(B)** Low magnification images of anti-LC3A/B staining in P17 retina. Scale bar = 20 μm. INL, inner nuclear layer; GCL, ganglion cell layer. **(C)** Immunoblot of P12 and 14 control and *Rabgef1*⁻/⁻ retinal lysates probed with antibody to LC3A/B (upper panel). Lower arrowhead indicates increase in the LC3A/B-II lipidated form in *Rabgef1*⁻/⁻ retinas. For quantification of LC3A/B-II level, P12 and P14 band intensities were combined and normalized to γ-tubulin (lower panel). **(D)** Left panels show low-magnification images of anti-p62 staining in P17 retina. Scale bar = 50 μm. Right panels show the same sections imaged at high magnification using Airyscan (AS*). Scale bar = 20 μm. **(E)** Transmission electron micrographs of *Rabgef1*⁻/⁻ photoreceptor inner segments at P17 immunolabeled with anti-p62/immunogold or immunogold without primary antibody. Arrowheads indicate anti-p62 immunolabeling in electron-dense deposits. Scale bar = 500 nm. **(F)** Immunoblot of P14 control and *Rabgef1*⁻/⁻ retinal lysates probed with antibody to p62. β-actin was used as a loading control.

analysis identified global expression changes as a prelude to photoreceptor degeneration, with a total of 2469 significant DE genes that could be categorized to specific cellular pathways including phototransduction, mitochondria, oxidative stress and endocytosis (Fig 7A). We then performed Gene Ontology (GO), KEGG and Reactome pathway analysis of DE genes (Fig 7B). Several genes associated with phototransduction and mitochondrial oxidative phosphorylation showed consistently lower expression at early stages (P6 and P10) of photoreceptor differentiation in the KO retina (Fig 7C). We note that the expression of these genes increases during normal development of photoreceptors [36]. We also observed altered expression of a number of oxidative stress response genes at the three stages examined (S7 Fig). Interestingly however, autophagy-associated genes were not differentially expressed in the transcriptome analysis (S8 Fig). Nevertheless, higher expression of lysosomal and early endosomal genes in the P14 *Rabgef1*-KO retina (Fig 7E) indicates an adaptive response to decreased protein recycling and degradation. The transcriptomic analysis of *Rabgef1*-KO retina thus highlights multiple cellular and developmental processes that are impacted by the absence of RabGEF1 before photoreceptor degeneration is evident after eye opening (P14).

## Discussion

Internalization of macromolecules and recycling or turnover of membrane components by endocytosis are critical for diverse cellular functions that maintain homeostasis. Rab5 and its regulators RabGEF1 and Rabaptin-5 are required for the biogenesis of early endosomes [41], which then deliver cargo macromolecules to recycling or degradation pathways [20]. In addition to promoting Rab5 activity as a GEF, RabGEF1 also binds ubiquitin [42] and its recruitment to the early endosome is controlled by ubiquitination [43]. Notably, the function of RabGEF1 has been investigated primarily in cell culture systems [32,42,44]. Only a few studies have been performed *in vivo*; specifically, the absence of RabGEF1 in mast cells and epidermal keratinocytes was demonstrated to contribute to skin inflammation in *Rabgef1*-KO mice [37,45]. Here, we implicate, for the first time, a critical function of RabGEF1 during photoreceptor maturation in the developing mammalian retina. The loss of RabGEF1 leads to severe defects in outer segment maintenance and phototransduction likely due to a decrease in protein recycling and membrane turnover caused by reduction in early endosomes marked by the Rab5-GTP effector protein EEA1. Concurrent to and/or followed by these events in developing photoreceptors, we demonstrate microscopically evident accumulation of macromolecular aggregates at P14 (eye opening) likely resulting from altered autophagy, which we propose leads to relatively fast photoreceptor cell death.

Developmentally regulated expression, concordant with photoreceptor morphogenesis [6] and significant loss of visual function in *Rabgef1*⁻/⁻ mice as early as eye opening (P14/P15), implied a major contribution of RabGEF1 to functional maturation and survival of retinal photoreceptors. We were intrigued by the photoreceptor-specific impact of RabGEF1 loss despite the ubiquitous nature of its function in endocytosis. Photoreceptor cell death was minimal at P14 and was only becoming evident by P21. Ectopic accumulation of outer segment proteins was not detected in *Rabgef1*⁻/⁻ photoreceptors; instead, slower biogenesis of outer segments appeared to result from reduced amount of OS bound proteins that reach their destination. Structural defects and reduced expression of phototransduction proteins can account only in part for disproportionately severe deficits in rod and cone function (both a- and b-wave) at P15 and P21. Dark rearing that slows photoreceptor degeneration in several retinal disease models (e.g., those with dominant rhodopsin mutations or with loss of the visual arrestin) did not have any impact on visual responses of *Rabgef1*⁻/⁻ mice. Another possible contributor to visual function defects could be a decrease in endocytosis leading to perturbation of

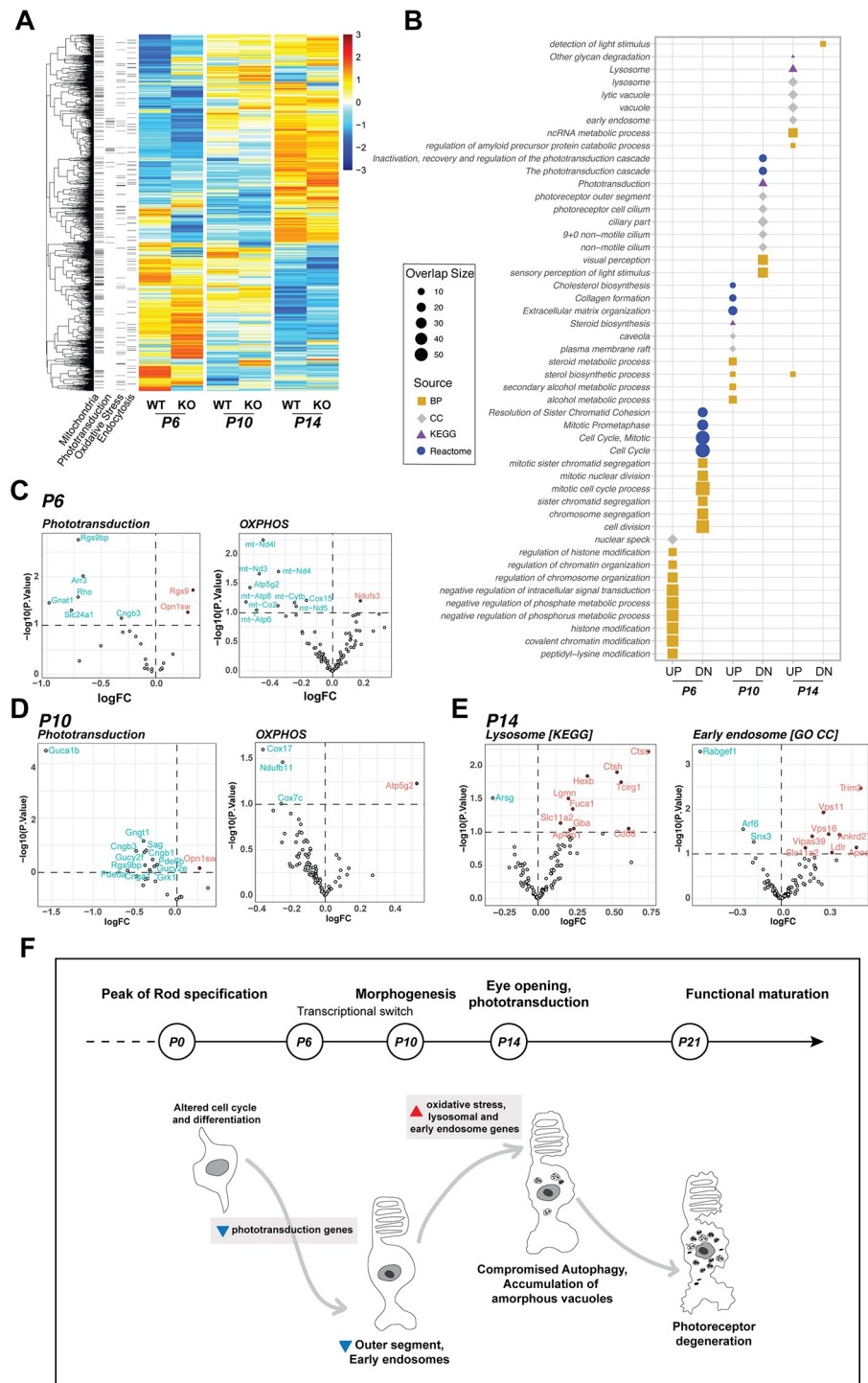

**Fig 7. Transcriptomic changes in the *Rabgef1*⁻ᐟ⁻ retina.** **(A)** Heatmap of differentially expressed genes in pairwise comparisons of age-matched control (WT) and *Rabgef1*⁻ᐟ⁻(KO) retina. The annotation block indicates whether differentially expressed genes are part of Mitochondria (Mitocarta), Phototransduction, Oxidative stress response pathway, and/or Endocytosis pathways. **(B)** Functional enrichment plot of differentially expressed genes for each time point. Colored shapes represent the source of pathways or gene groups: Gene Ontology Biological Processes (BP; gold square), Gene Ontology Cellular Component (CC; grey diamond), KEGG (purple triangle), or Reactome (blue circle). Only top 10 most significant (by p-value) pathways are plotted. **(C)** Early changes in *Rabgef1*⁻ᐟ⁻ retina at P6. Volcano plots show differential expression of phototransduction and OXPHOS genes at P6, soon after rod photoreceptor birth. Red and blue labels in volcano plots denote gene over- and under- expression with significance, respectively. **(D)**

Volcano plots showing continued downregulation of phototransduction and OXPHOS genes in *Rabgef1*<sup>-/-</sup> retina at P10. **(E)** Volcano plots showing significant upregulation of lysosomal and early endosome related genes in *Rabgef1*<sup>-/-</sup> retina at P14. **(F)** A possible model of photoreceptor cell death in mouse retina lacking RabGEF1, illustrating the centrality of its function in endocytosis and autophagy during normal development and homeostasis.

synaptic transmission from photoreceptors to bipolar neurons. Rab5 is essential for synaptic vesicle endocytosis and recycling [46]. A quantifiable and significant reduction in EEA1-positive puncta (without a change in EEA1 protein itself) representing early endosomes in *Rabgef1*<sup>-/-</sup> photoreceptor inner segments is consistent with decreased endocytosis upon knockdown of Rab5 isoforms [41]. The almost 3-fold decrease in several synaptic proteins in the vesicle fraction of KO retina (see S2 Table) is likely reflective of reduced endocytosis as well.

Reduced endocytosis, however, cannot satisfactorily explain the relatively fast death of photoreceptors in the absence of RabGEF1. We were intrigued by the accumulation of large amorphous electron-dense material within autophagosome-like structures observed upon ultrastructural analysis of *Rabgef1*<sup>-/-</sup> photoreceptors as early as eye opening in P14 retina. We wondered whether RabGEF1 loss leads to dysregulation of autophagy in photoreceptors. Both endocytosis and autophagy involve fusion of membrane vesicles to lysosomes for degradation and utilize many similar components associated with cell signaling and trafficking [47,48]; this convergence has implicated a regulatory crosstalk between these two evolutionarily conserved transport pathways that control cellular homeostasis. Rab5 directly regulates autophagosome formation in the canonical autophagy pathway [39,49]. However, its association with the autophagy marker LC3 is also implicated in endocytic removal of β-amyloid in a mouse Alzheimer's disease model [50]. Notably, siRNA-mediated silencing of *Rabgef1* is shown to result in enhanced autophagy in *C. elegans* [51] and the involvement of the ubiquitin-binding domain is suggested to be responsible for RabGEF1's ability to switch back and forth between endocytic and autophagic functions [52]. Our results demonstrating enhanced LC3 immunoreactivity in *Rabgef1*<sup>-/-</sup> photoreceptors strongly implicate a direct role of Rab5 and RabGEF1 in maintaining normal autophagic homeostasis [53].

At present, we cannot conclude whether the autophagy phenotype of the *Rabgef1*<sup>-/-</sup> photoreceptors reflects increased autophagy initiation or decreased autophagy flux. A distinction between these two possibilities would require pharmacologic manipulations (*e.g.*, bafilomycin A1 treatment) that are only feasible in cell culture systems. Increased autophagy initiation could be a response to the accumulation of internalized proteins that cannot recycle back to the plasma membrane or traffic to lysosomes via late endosomes. This interpretation would be in line with the enhanced autophagy observed upon silencing of *Rabgef1* in *C. elegans* [51]. On the other hand, decreased autophagic flux could result from the involvement of Rab5 and RabGEF1 in autophagosome biogenesis [39,49]. The increased immunostaining of p62-positive aggregates in *Rabgef1*<sup>-/-</sup> photoreceptors observed in our studies would be most consistent with this latter possibility, since p62 itself is degraded by autophagy. We note that aggregates of p62 and LC3 visualized by immunofluorescence staining in the *Rabgef1*<sup>-/-</sup> retina were detected only in a subset of photoreceptors and that the immunoblot analysis of p62 and LC3 protein levels was performed with total retinal extracts. p62 and LC3 aggregates in *Rabgef1*<sup>-/-</sup> photoreceptors reflect undigested autophagosomes, and the total amount of the two proteins might not be dramatically altered in the whole retina. Further autophagic flux experiments will be required to elucidate the underlying defects in the process. In any event, we hypothesize that reduced endocytic trafficking and accumulation of membrane components in autophagic vesicles disrupts normal physiological functions leading to death of photoreceptor cells (Fig 7F).

Consistent with this hypothesis, temporal transcriptomic profiling of KO retina reveals early changes in the gene expression associated with phototransduction and ciliary proteins resulting in aberrant photoreceptor maturation. We also uncovered dysregulation of genes associated with mitochondria, which could contribute to enhanced oxidative stress in photoreceptors prior to cell death. Augmented expression of endosomal and lysosomal genes is likely indicative of a photoreceptors' compensatory response in an attempt to eliminate accumulated protein aggregates. We note that disruption of the autophagy-lysosomal pathway followed by abnormal mitochondrial turnover in rod photoreceptors has been suggested to cause early retinal degeneration in mice carrying a mutation in the *Cln6* gene (*Cln6^{nclf}*) [54].

In conclusion, our studies uncover a direct and critical role of RabGEF1 in the endocytic and autophagic pathways during functional maturation of mammalian photoreceptors and demonstrate that the loss of RabGEF1 leads to photoreceptor developmental defects and relatively fast retinal degeneration likely because of compromised recycling and proteostasis. We suggest that *Rabgef1* and other endocytosis/autophagy-associated effector genes should be considered as candidates for syndromic diseases that include photoreceptor dysfunction or degeneration.

## Materials and methods

### Ethics statement

All procedures involving the use of mice were approved by the Animal Care and Use Committee of the National Eye Institute (ASP #650) and performed in accordance with the Statement for the Use of Animals in Ophthalmic and Vision Research of the Association for Research in Vision and Ophthalmology.

### Animals

BALB/cJ mice were purchased from Charles River Laboratories (Germantown, MD, USA). *Rabgef1^{-/-}* mice have been described previously [37]. *Rabgef1^{+/-}* males (from Dr. S. Galli, Stanford University) were crossed to wildtype BALB/cJ females to refresh the colony. We used intercrosses of heterozygous mice (KO mice were not viable on BL/6 background) after genotyping using the following: shared primer (6505) 5'- TTA GAG GAG GTT GTG AGC TGCCAT-3'; wildtype primer (6506) 5'- TGC AGC TTA CTC AGG CAT GGA AGA-3'; and mutant primer (6534) 5'-GAC GTG CTA CTT CCA TTT GTC ACG -3'. The PCR was performed as follows: Step 1. 94˚C for 60 sec. Step 2. 94˚C for 15 sec. Step 3. 66˚C for 60 sec. Step 4. 72˚C 90 sec. Repeat steps 2, 3, and 4 for 35–40 cycles, depending on concentration of starting DNA. Step 5. 72˚C 10 min. Step 6. 4˚C. It should be noted that the *Rabgef1* PCR reaction required 0.5M Betaine (Sigma) and 3% DMSO by volume, in order to stabilize the DNA within the reaction. The mutant band is 1 kb in size, while the wildtype is 840 bp. Original founders were negative for *Rd1* and *Rd10* mutant alleles that are sometimes present within inbred mouse strains. Male and female mice were used equally in all experimental procedures and maintained on a normal 12 h light/dark cycle (unless otherwise noted) with unlimited access to food and water. Both heterozygous and wildtype littermates were used as controls.

### Electroretinogram (ERG) recordings

The mice were dark-adapted overnight and prepared for recording under dim-red illumination. The pupils were dilated by topical administration of tropicamide (1% wt/vol, Alcon) and phenylephrine (2.5% wt/vol, Alcon). Proparacaine hydrochloride (0.5% wt/vol, Alcon) was applied for topical eye anesthesia. Mice were anesthetized with ketamine (100 mg/kg) and

xylazine (10 mg/kg) by I.P. injection of 0.1 mL/10 g body weight of anesthetic solution (1 mL of 100 mg/mL ketamine and 0.1 mL of 100 mg/mL xylazine in 8.9 mL 1X PBS). Body temperature was maintained at 37°C on a heating platform. Gonioscopic Prism solution (2.5% wt/vol, Hypomellose Ophthalmic Demulcent solution, Alcon) was applied to each eye, and gold wire loop electrodes were placed on corneas, with the reference electrode placed in the mouth. Flash ERG recordings were taken simultaneously from both eyes using an Espion E2 Visual Electrophysiological System (Diagnosys). Rod and mixed rod-cone ERG responses were recorded at increasing light intensities over the range of 0.0001–10 cd·s/m$^2$ under dark adapted conditions. The stimulus interval between flashes varied from 5 s at the lowest stimulus strengths to 60 s at the highest ones. Two to ten responses were averaged depending on flash intensity. The mouse was then light adapted for photopic responses for 2 minutes in the Ganzfeld dome. Recordings were carried out at light intensities over 0.3–100 cd·s/m$^2$ under a background light that saturates rod function. Twenty to 30 responses were averaged. ERG signals were sampled at 1000 Hz and recorded with a 0.3 Hz low-frequency and 300-Hz high frequency cutoffs. The a-wave amplitude was measured from the baseline to the negative peak, and the b-wave was measured from the a-wave trough to the maximum positive peak. To compare ERG data between control and *Rabgef1$^{-/-}$* animals at each intensity, we first averaged the ERG amplitude data for the two eyes of each mouse and then averaged the data for all mice within a single group (e.g., genotype). Statistical values were calculated using Prism Software with one two-tailed unpaired t-test performed per row. P values were computed with more power, assuming that all rows are sampled from populations with the same scatter (SD). Multiple comparisons were corrected for using the Holm-Sidak method to calculate statistical significance. * (0.033), ** (0.002), *** (<0.001).

## Histology and transmission electron microscopy (TEM)

For standard histology, mice were euthanized, the anterior segment was removed before fixing eyecups in 4% glutaraldehyde in 1X PBS for 30 min. Eyecups were then transferred to 4% PFA in 1X PBS until being embedded in glycol methylacrylate. Sections were cut 5-micron thick, stained with hematoxylin and eosin, and imaged using a Zeiss Axio Imager Z1 brightfield microscope. For TEM, eyes were fixed in PBS-buffered glutaraldehyde (2.5% at pH 7.4) and PBS-buffered osmium tetroxide (0.5%) and embedded in epoxy resin. Thin 90 nm sections were collected on 200-mesh copper grids, dried for 24 h, and double stained with uranyl acetate and lead citrate. Sections were viewed and photographed using a JEOL JM-1010 electron microscope.

For TEM immunolabeling, eyes were enucleated, incised at the limbus and fixed by immersion in cold 2% paraformaldehyde in 1X PBS for 1 hour. Anterior segment and lens were dissected. Eye cups were dehydrated in a graded ethanol series (30% - 85% ethanol) on ice. Eye cups were transferred to LR White Resin (medium grade) (Electron Microscopy Sciences, Hatfield PA) for 1 hr then transferred to fresh LR White resin and infiltrated for 18 hours at 4 C. Eye cups were transferred to embedding molds containing fresh LR White and air was evacuated from molds to facilitate polymerization. The resin was polymerized for 24 hours at 60 C. A Leica UC6 ultramicrotome and diamond knife were used to cut thin sections which were collected on nickel grids. Sections were blocked in 5% normal goat serum diluted in EM ICC buffer (20mM Tris, 135 mM NaCl, 0.1% BSA, 0.05% Tween-20 pH 7.3) and immunolabeled with rabbit anti-P62 antibody (Abcam Cat #109012) at 8.6ug/ml diluted in EM ICC buffer for 2 hr at room temperature. Primary antibody was omitted from negative control samples. Grids were washed repeatedly and incubated in goat anti-rabbit 20nm gold (Abcam Cat # 27237) diluted 1:20 in EM ICC buffer for 1 hr, washed repeated and fixed for 5 min in 2% glutaraldehyde, washed in distilled water and allowed to dry. Grids were stained with 1% uranyl acetate

for 18 minutes and Reynold's lead citrate for 5 minutes. Grids were imaged on either a JEOL 1010 or JEOL 1200 EX II.

## Immunoblot and immunoprecipitation

Immunoblotting was performed as described [55] using antibodies listed in S3 Table. Images are representative of at least three biological replicates. Densitometric analysis was performed using either ImageJ or Image Lab (BioRad) by normalizing proteins to β-actin, γ-tubulin, or total protein. Statistical significance was determined using an unpaired t-test within the Prism Program.

The Dynabeads (Thermo Scientific, Waltham, MA) protocol was used for immunoprecipitation studies. Briefly, 5μg Rabex5/RabGEF1 antibody (Sigma-Aldrich, St. Louis, MO) was covalently conjugated to 1 mg of Dynabeads and incubated overnight with retina extracts at 4°C. After removing the supernatant, beads were washed with cold immunoprecipitation buffer (150 mM NaCl, 2 mM EDTA, 40 mM Tris-HCL pH 7.4, 1% NP-40, 10% Glycerol with 1X Complete Protease Inhibitors (Roche, Basel, Switzerland) three times for 10 min each time. Beads were then washed with Dynabeads reagent, LWB, 3X for 10 mins minimum each time. Bound proteins were eluted in 20–40 μl of EB (elution buffer) from the Dynabeads kit and subjected to SDS-PAGE and immunoblotting.

## Immunofluorescence analysis

Whole eyes were either frozen immediately in optimal cutting temperature (OCT) solution (Tissue Tek, Sakura, USA) (for fresh-frozen sections) or fixed in 4% paraformaldehyde (PFA) for 30–60 min, followed by overnight cryoprotection in 30% sucrose in 1X PBS, and then embedded in OCT. The anterior segment was dissected from fixed eyes prior to embedding. Fresh frozen cryosections were post-fixed for 2–5 min with 4% PFA, and all cryosections were permeabilized and blocked with 5% Donkey Serum in 1X PBS with 0.3% Triton X-100 for at least 30 min at room temperature before incubation with primary antibodies overnight at 4°C. Primary antibody concentrations and reaction conditions can be found in S3 Table. After antibody incubations, slides were washed for 30–60 min at room temperature with 1X PBS, mounted using Southern Biotech Fluoromount media (Birmingham, AL, USA), and viewed on a confocal laser scanning microscope unless otherwise indicated (Zeiss, Model 700 or 880 with Airyscan, Jena, Germany).

## EEA1 (early endosomal antigen 1) quantitation in photoreceptors

Confocal images of EEA1-stained retina sections were collected in 4 slice Z-stacks with an interval of 0.5 μm, thus sampling from 2μM thickness of tissue. Image stacks were converted to grayscale and cropped to the photoreceptor inner/outer segment area where the bulk of the photoreceptor cellular machinery is located. EEA1-positive puncta were counted from individual slices using ImageJ.

## *In Situ* hybridization

*In situ* hybridization was performed according to manufacturer's instructions [Advanced Cell Diagnostic Biosystems (ACD), fluorescence protocol; Newark, CA, USA]. Specific probes were designed by ACD against the 3' and 5' untranslated regions of *Rabgef1* transcript.

## Preparation of vesicle fractions from mouse retina

Vesicle fractions enriched for autophagosomes were isolated from retinas of *Rabgef1*-KO and control mice using density gradient centrifugation, as described [56] with modifications

suggested by Dr. S. Azadi. Briefly, 20 retinas of P14-P19 mice were homogenized in 500 μl of cold HEPES buffer (50mM each of HEPES, TES, Tricine, 100 mM NaCl, 10mM KCl, 1mM each of $KH_2PO_4$ and $Na_2SO4$, 5mM $MgCl_2$, 1mM $CaCl_2$) pH 7.6 using a motorized pestle. After adding additional 500 μl of cold HEPES buffer, the suspension was passed 10 times through a 20G needle and incubated at 37˚C for 3 h with 50 μM Vinblastine. This step blocks the translocation of autophagosomes to lysosomes. The sample was cooled and diluted three times with a hypotonic buffer (10mM HEPES, 1 mM EDTA, pH 7.4) to lyse the cells and centrifuged for 5 min at 1400xg to remove nuclei and cell debris. The supernatant was treated with 0.5 mM of GPN (Glycyl-L-Phenylalanyl 2-naphthylamine) at 37˚C for 10 min to disrupt and break the lysosomes. The sample was then centrifuged at 4000xg for 2 min to remove the remaining nuclear debris. The supernatant was loaded on top of a discontinuous Nycodenz gradient (2.9 ml of 22.5% at the bottom/ 6.7 ml of 9.5% in the middle/ 3.1 ml of the supernatant sample on the top) and centrifuged for 1 h at 28,000xg at 4 degrees. Two bands were collected, one at the interface between the sample and 9.5% Nycodez, and the other at the 9.5% and 22.5% interphase. The fractions containing partially purified autophagosomes were analyzed by immunoblotting and subjected to mass spectrometry.

## Liquid chromatography-mass spectrometry analysis

For proteome analysis, immunoprecipitated proteins were reduced, alkylated and trypsin digested as previously described [57]. Desalted tryptic peptides were analyzed on an Orbitrap Lumos Tribrid mass spectrometer coupled with a UltiMate 3000-nLC (Thermo Fisher Scientific). Raw data files were processed with Proteome Discoverer (v2.2, Thermo Fisher Scientific) software, using Mascot v2.5.1 (Matrix Sciences) search node for peptide/protein identification. Target Decoy was used to calculate the false discovery rate (FDR) of peptide spectrum matches, set to a p-value <0.05 [58]. Geneset enrichment analysis with fold change ratio of peptide spectral match counts between KO and WT retina was performed as described in the RNA seq analyses section below.

## RNA-seq analyses

Total retinal RNA was used for preparing libraries for RNA-seq, and the data was analyzed using *Mus musculus* annotations v84 from Ensembl as described [30,59]. Gene abundance were processed with edgeR and limma in R for normalization and differential expression tests between age-matched WT and *Rabgef1*-KO samples. A standard p-value cut off of 0.1 was used to qualify differentially expressed genes. Genelist enrichment was done with the gProfileR (v0.6.7) [60] package, whereas gene set enrichment analysis was performed using fgsea (v1.8.0) [61]. Reference pathway annotations for geneset enrichment analysis were obtained from ConsensusPathDB [62], GSKB (http://ge-lab.org/gskb/) and manually curated genelists for selected pathways, as described [30]. Additionally, R packages (tidyverse, pheatmap, eulerr and ggrepel) were used for analysis and visualization with custom scripts.

## Supporting information

**S1 Fig.** (A) Total protein loading control (left panel) for RabGEF1 immunoblot shown in Fig 1D, and quantification of total RabGEF1 expression (right panel). (B) Quantification of outer nuclear layer (ONL) thickness and inner and outer segment (IS-OS) length from hematoxylin and eosin stained sections.
(TIF)

**S2 Fig. Immunohistochemistry of control (Ctrl) and *Rabgef1*<sup>-/-</sup> (KO) littermate retinas at P10, P12 and P21 using cell type-specific antibody markers.** Rho (Rhodopsin, a marker of rods), Cone Arr (Cone arrestin, a marker of cones), PKC-α (a marker of ON-bipolar cells), Calbindin (a marker of Horizontal, Amacrine, and some Ganglion cells), GFAP (glial fibrillary acidic protein, marker of Müller glia stress), and Ribeye (a marker of photoreceptor ribbon synapses). Retinal sections were counterstained with DAPI to visualize nuclei. OS, outer segment; IS, inner segment; ONL, outer nuclear layer; OPL, outer plexiform layer; INL, inner nuclear layer; IPL, inner plexiform layer; GCL, ganglion cell layer. Scale bar = 20 μm.
(TIF)

**S3 Fig. Raw electroretinogram (ERG) traces from Rabgef1-KO and control animals.** Representative traces are shown at increasing light intensities for pairs of control and Rabgef1-KO animals. Reduced function is indicated by smaller a-wave (first negative peak) and b-wave (positive peak). The a- and b-wave peaks are labelled in the top left trace for reference.
(TIF)

**S4 Fig. Additional replicate transmission electron micrographs of *Rabgef1*<sup>-/-</sup> photoreceptors from P10 to P22 showing accumulation of amorphous proteolytic aggregates as development proceeds.** Scale bar = 2 μm.
(TIF)

**S5 Fig. Characterization of the Rabgef1<sup>-/-</sup> retina.** (A) Expression of Rab5 protein in Control (Con) and Rabgef1-KO (KO) animals at P14. (B) Representative EEA1 immunohistochemistry images used for quantification shown in Fig 5C and 5D. (C) Immunohistochemistry of anti-LC3A/B on Rhodopsin T17M mutant 6-week-old littermates. There is no visible accumulation of autophagosomes marked by LC3A/B. Retinal sections were counterstained with DAPI. ONL, outer nuclear layer; INL, inner nuclear layer. Scale bar = 20 μm. (D) TEM-acquired images of *Rd1* and *Rds* mouse models of retinal degeneration.
(TIF)

**S6 Fig. Altered vesicular cargo composition in enriched fractions of *Rabgef1-KO* retina.** (A) Summary of proteins identified from mass spectrometric analysis of vesicle enriched subcellular fractions from control and *Rabgef1*<sup>-/-</sup> retina. The Venn diagram also shows how many of the detected proteins are annotated in endocytosis or autophagy pathways. (B) Cargo proteins enriched in control samples, ranked by decreasing order of enrichment (logFC Control/ *Rabgef1*<sup>-/-</sup>). Proteins of Autophagy, Endocytosis, Mitocarta [64], Synapse and Phototransduction are highlighted in specific colors as described in the legend, or colored black if annotated in multiple categories. Proteins not belonging to the aforementioned groups are colored grey. (C) Enrichment plots for significant under enrichment of Phototransduction and Mitochondrial outer membrane gene sets as observed in gene set enrichment analyses of vesicular cargo from control and *Rabgef1*<sup>-/-</sup> retinas.
(TIF)

**S7 Fig. Dysregulation of oxidative stress response genes in the *Rabgef1-KO* retina.** Volcano plots summarize significant differential expression of genes associated with the oxidative stress response pathway at P6, P10, and P14. Each point represents a gene where labelled or colored red and blue denote significant over-expression and under-expression in Rabgef1-KO vs WT comparisons, respectively.
(TIF)

**S8 Fig. Transcriptional status of autophagy associated genes at different timepoints preceding *Rabgef1*<sup>-/-</sup> related degeneration.** Genes participating in distinct stages of autophagy,

namely: Turning on the pathway, Initiation of autophagosome biogenesis, Building the autophagosome, and Fusing autophagosome with lysosome, were obtained from [65], and investigated for their activity in the *Rabgef1-KO* retina. Volcano plots are arranged in a grid of time versus stage of autophagy, and a point in the volcano plots represents a single gene where colors red and blue denote significant over-expression and under-expression in *Rabgef1*-KO vs WT comparisons, respectively.
(TIF)

**S1 Table. RNA-expression of Rab family genes in mammalian retina (E11-P28) and isolated flow-sorted photoreceptors (P2-P8).** Gene-level quantification in counts per million (CPM; average of biological replicates) of Rabs and associated GEFs, GAPs and co-factors in the mouse retina, and flow sorted photoreceptors (FSPR)—rod and S-cone-like. Rabgef1 is highlighted and in bold to indicate its characteristic gene expression pattern during photoreceptor development. Data was extracted from Kim et al. 2016.
(XLSX)

**S2 Table. Mass spectrometry based proteomic characterization of endosomal vesicular fractions from control and *Rabgef1*$^{-/-}$ retina (n = 20 pooled P14-P19 retinas per genotype).** All identified protein species are listed with peptide spectral match (PSM) counts from control and *Rabgef1*$^{-/-}$ samples. Fold change is calculated by dividing PSM counts from *Rabgef1*$^{-/-}$ samples to controls.
(XLSX)

**S3 Table. Resource table listing all antibodies used within the study.**
(XLSX)

## Acknowledgments

We are grateful to Drs. Mindy Tsai, See-Ying Tam and Stephen Galli at Stanford University for providing the *Rabgef1*$^{+/-}$ mice and for technical advice during the study, to Megan Kopera and the NEI Genome Engineering Core for re-derivation of the *Rabgef1*$^{-/-}$ line. We acknowledge Ximena Corso-Diaz, Thad Whitaker and Christie Campla for technical advice. We thank Dr. Seifolla Azadi for retinal vesicle isolation protocol, Dr. Christopher K. E. Bleck, NHLBI Electron Microscopy Core and Dr. Mones Abu-Asab, NEI Electron Microscopy Core for access to TEM facilities. This study utilized the high-performance computational capabilities of the Biowulf Linux cluster at National Institutes of Health (http://biowulf.nih.gov).

## Author Contributions

**Conceptualization:** Passley Hargrove-Grimes, Anand Swaroop.

**Data curation:** Anupam K. Mondal.

**Formal analysis:** Passley Hargrove-Grimes, Anupam K. Mondal, Haohua Qian, Juan S. Bonifacino, Tiansen Li, Anand Swaroop.

**Funding acquisition:** Anand Swaroop.

**Investigation:** Passley Hargrove-Grimes, Jessica Gumerson.

**Methodology:** Passley Hargrove-Grimes, Jessica Gumerson, Jacob Nellissery, Angel M. Aponte, Linn Gieser, Robert N. Fariss, Tiansen Li.

**Project administration:** Anand Swaroop.

**Resources:** Jacob Nellissery, Angel M. Aponte, Haohua Qian, Robert N. Fariss, Juan S. Bonifacino, Anand Swaroop.

**Software:** Anupam K. Mondal.

**Supervision:** Tiansen Li, Anand Swaroop.

**Validation:** Jessica Gumerson, Jacob Nellissery.

**Visualization:** Passley Hargrove-Grimes, Anupam K. Mondal, Jessica Gumerson.

**Writing – original draft:** Passley Hargrove-Grimes, Anupam K. Mondal, Anand Swaroop.

**Writing – review & editing:** Passley Hargrove-Grimes, Anupam K. Mondal, Jessica Gumerson, Jacob Nellissery, Angel M. Aponte, Haohua Qian, Robert N. Fariss, Juan S. Bonifacino, Tiansen Li, Anand Swaroop.

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
