## [Decision Letter · Decision Letter 0]

8 May 2020

Dear Anand, 

Thank you very much for submitting your Research Article entitled 'Loss of endocytosis-associated guanine nucleotide exchange factor RABGEF1 causes defects in photoreceptor outer segment biogenesis leading to retinal degeneration' to PLOS Genetics.

Our apologies for the delay in reaching a decision, which was due to difficulty securing reviewers, compounded (as you know) by an unfortunate lapse on the part of PLOS to track important manuscript correspondence. Although the lapse took place in the middle of the pandemic, that is not a sufficient excuse, and we apologize (again) for the delay.

The manuscript was fully evaluated at the editorial level and by three external peer reviewers. The reviewers' recommendations are minor revision, reject, and major revision. As you will see, while the reviewers find the work interesting, they express substantial concerns that cast doubt on the strength of the novel conclusions that can be drawn at this stage. Rab5 should be the main target for RABGEF1/Rabex5, but data on rab5 in the retina and especially photoreceptor cells are lacking in the present study. While Rab5 functions are well known for the RPE related to the phagocytosis machinery for Rod outer segments, neither the expression profile nor the function of rab5 the vertebrate neuronal retina or photoreceptor cells has been established. Therefore, it seems questionable whether the available knowledge about the effects of the absence of RABGEF1 really originates from photoreceptor cells, since the OMICs data are from the entire retina. In any case, additional functional data are necessary to enlighten the link between RABGEF1-related defective endosome function and autophagy in the present retinal degeneration model.

Based on the reviews, we will not be able to accept this version of the manuscript, but we would be willing to review again a much-revised version. We cannot, of course, promise publication at that time.

If you decide to revise the manuscript for further consideration at PLOS Genetics, please aim to resubmit within the next 60 days, unless it will take extra time to address the concerns of the reviewers, in which case we would appreciate an expected resubmission date by email to plosgenetics@plos.org.

We are sorry that we cannot be more positive about your manuscript at this stage. Please do not hesitate to contact us if you have any concerns or questions.

Yours sincerely,

Uwe Wolfrum

Guest Editor

PLOS Genetics

Gregory Barsh

Editor-in-Chief

PLOS Genetics

**Comments to the Authors:**

Reviewer #1: This authoritatively-written manuscript, presents evidence that depletion of Rabgef1 adversely affects photoreceptor outer segments and leads to retinal degeneration in murine mutants. Analysis of transcriptome data implicated Rabgef1 as a candidate and via experiments that extend from expression analysis, histology, electroretinography and immunohistochemistry, to TEM, IPs, MS etc., the authors demonstrate that Rabgef1 loss induces profound outer retinal phenotypes. Notably, these data are presented in a thoughtful and linear manner. The results are important and are expected to represent the foundation for future studies investigating the role of Rabgef1 in ocular disease.

I was unable to discern any scientific weaknesses to the data presented, and offer the following minor suggestions for textual revisions, that may enhance the publication.

1. Scope exists to enhance the background regarding the roles of Rabs, both by including a diagram/cartoon illustrating the inter-relationship of Rabs, Gefs and Gaps; and by earlier description of paralogs that induce retinal dystrophies. Although Rab27 and 28 are mentioned in the final paragraph of the discussion, more comprehensive review of this gene network would be beneficial for PLoS Genetics’s diverse readership.

2. Although Rabgef1 heterozygous mice likely represent the closest model for human disease, their phenotype is unreported. Inclusion of a description would be of value, unless this would complicate a follow up study focusing on the human genetics of Rabgef1 mutation.

Reviewer #2: Using transcriptomics data, the authors have identified Rabgef1 as the only rab-associated protein that is upregulated during photoreceptor development. Given the importance of Rab function in photoreceptor function, they studied a rabgef1 knockout mouse. They find that rabgef1 knockout mice undergo retinal degeneration, although the mechanism of this retinal degeneration remains unclear. Since rabgef1 is a GEF for rab5, they suggest a mechanism related to a reduction in rab5 activity, which is supported by the reduction in EEA1 punctae that they claim to observe, however they don’t actually present any data on rab5 in their study. Furthermore, the link between the apparent reduction in EEA1 punctae and the interesting observation of LC3-labeled punctae and EM autophagosomes is unclear. The transcriptomic and proteomic data presented in figures 6 and 7 do not help to clarify the story. Overall, the rabgef1 knockout phenotype is interesting and the observation of increase LCA3-labeled punctae is also very interesting, but the story seems to stop at a descriptive and correlative level, and thus feels incomplete.

A general weakness is the use of whole retinal lysate for various experiments to test the specific role of RABGEF1 in photoreceptor development, despite its wide expression pattern in many retinal cell types. The RNA-seq data also stem from total retinal RNA, thus hindering specific relevance to photoreceptors.

Specific points:

Page 7, paragraph 2-It would be useful to have some quantification of the photoreceptor phenotype, possibly an actual measurement of outer segment length.

Page 8, paragraph 2 (Figure 5)-The authors claim there was much less EEA1 immunoreactivity in the KO than in the control. It would be useful for the authors to mention how this quantification was done. If it was performed on a single Z-slice, as is suggested, was the same Z-slice used for both conditions, and how was it chosen? Is it possible that the knockout has EEA1 in a different retinal slice than the wild type? Also it would be useful for the authors to show the full western blot from Figure 5E.

Figure 6-The increase in LC3A/B-labeled punctae in the knockout is striking. Perhaps the authors could show the full western blot from Figure 6C, to confirm the specificity of their antibody?

Page 9, paragraph 2-the authors looked at endosomal vesicular fractions, which they argued should be enriched in autophagosomes, and observed a decrease in phototransduction proteins and mitochondrial homeostasis proteins in their Rabgef1 knockouts relative to the controls. If there were an increase in autophagosomes, wouldn’t you expect the opposite? But then again, does this quantification really mean anything, given the decrease in number of photoreceptors in the knockout? At what age was the quantification performed?

Overall, the proteomic and transcriptomic analysis in Figure 6 and 7 don’t seem to add much to the paper, and don’t really help to explain the phenotype. The most striking result in the study is the increase in LC3A/B labeled punctae in Figure 6. It would be interesting to know, however, whether the objects, claimed to be autophagic structures (Fig. 4), can be labeled with an autophagic marker, such as LC3.

Reviewer #3: On early endosomes, activation of Rab5 is catalyzed by the GEF Rabex5 complexed with the effector Rabaptin5 contributing to early endosome generation and trafficking.

In the current paper, with the title ‚Loss of endocytosis-associated guanine nucleotide exchange factor RABGEF1 causes defects in photoreceptor outer segment biogenesis leading to retinal degeneration‘, Hargrove-Grimes et al show, that loss of Rabgef1/Rabex5 in mice resulted in a failure of sufficient early endosome production and endosome trafficking leading to an upregulation of autophagy in the retina and defects of retinal photoreceptor morphology. Using the RABGEF1 KO mouse, the authors show that RABGEF1 is necessary for rod photoreceptor function and maintenance. As cause for progressive degeneration starting after P14, the authors discuss defects in endosome formation leading to an increase in autophagosome formation suggesting autophagy malfunction.

The study suggests an essential role of the endocytic machinery in photoreceptor outer segment biogenesis and function as absence of Rabex5 resulted in an almost complete loss of both rod and cone function early on.

Thes study is shedding light on the importance of the endocytic machinery in delivering cargo for ciliary trafficking of photoreceptors towards outer segment generation. As such I regard it as principally well suited to be published in PLoS Genetics.

There are some criticisms which should be addressed before publication:

A general shortcoming of the study is its focus on rod photoreceptors, disregarding the cones. The manuscript would greatly increase significance, if the cone phenotype would be analyzed after loss of Rabgef1/Rabex5 in mice. Please include data cone opsin trafficking and cone behavior. Is for example cone death a primary event or a consequence of rod death (secondary cone degeneration)

Figure 3 shows the reduction of rhodopsin but there is no data on cone opsins. What is the situation there, given that both a and b wave are greatly reduced?

Figure 1B: A negative control either using a sense probe or the P21 KO would be helpful to distinguish between background and specific staining.

Figure 3B and corresponding text: The blots suggest a delay in protein expression, which does not fit to the discussed ‘decreased recycling of membrane proteins’. In the transcriptome dataset a downregulation of phototransduction genes. How do the authors explain/discuss this early effect of RABGEF1-KO, which happens before photoreceptor-specific RABGEF1 expression is upregulated?

Figure 5C: In P10 samples, more EEA1 staining seems to be in the ONL, which was not quantified. Is there a mis-localization of EEA1-positive endosomes which results in less positive endosomes in the IS? It would be good to highlight the area taken for measurements for Figure 5D. A close-up of the selected area showing single EEA1-positive endosomes is needed to strengthen data shown in 5D. In the Figure legend title there is a typo with RABGEG1 instead of RABGEF1.

Suppl. Table S1 and S2: Please state which stages were used (P21?) in the legend.

Rabex5/RABGEF1 and other components of the endocytic pathway may however, be as essential for overall development of a vertebrate organisms, that they may not show up as potential candidates for diseases that include retinopathy phenotypes.

Suppl. Table S2, Figure 5B and Figure 6D-F and corresponding methods section: How many replicates did you measure? If only one, at least two more biological replicates should be prepared and statistically validated.

Figure 6D-F: How did you perform the isolation of vesicular fractions? This part is missing in the method section.

**Have all data underlying the figures and results presented in the manuscript been provided?**

Reviewer #1: Yes

Reviewer #2: Yes

Reviewer #3: Yes

PLOS authors have the option to publish the peer review history of their article (what does this mean?). If published, this will include your full peer review and any attached files.

Reviewer #1: No

Reviewer #2: No

Reviewer #3: No

---

## [Decision Letter · Decision Letter 1]

29 Oct 2020

Dear Anand,

Thank you very much for submitting your Research Article entitled 'Loss of endocytosis-associated guanine nucleotide exchange factor RabGEF1 causes aberrant morphogenesis and altered autophagy in developing photoreceptors leading to retinal degeneration' to PLOS Genetics. Your manuscript was fully evaluated at the editorial level and by independent peer reviewers. The reviewers appreciated the attention to an important topic but identified some aspects of the manuscript that should be improved.

We therefore ask you to modify the manuscript according to the review recommendations before we can consider your manuscript for acceptance. Your revisions should address the specific points made by each reviewer.

[LINK]

Yours sincerely,

Uwe Wolfrum

Guest Editor

PLOS Genetics

Gregory Barsh

Editor-in-Chief

PLOS Genetics

The authors have addressed the minor points from reviewer and most of the critical concerns from reviewer 2 and 3 the reviewers. Reviewer 2 accepted to review the revised version and states that most of her/his concerns can been ruled out, now and that the revised manuscript has very much improved. I agree with this and would like to extend this to the criticisms of the 3rd reviewer, who insisted on a major revision, but unfortunately did not re-examine the revised manuscript.

The 3rd reviewer asked for further comments on the cone phenotype in Rabgef1-/- mice. The authors added immunostaining for cone arrestin in a supplement figure, but do not mention or comment the results in the text of the manuscript. I suggest comment the findings on cone staining in a revised version.

I also agree that the new title reflects their findings better. Nevertheless, I suggest shortening of the title.

Reviewer's Responses to Questions

**Comments to the Authors:**

Reviewer #2: Comments:

The authors have addressed some of the critical concerns, and the paper is very much improved.

I agree with the main conclusions that 1) Rabgef1KO leads to photoreceptor defects. 2) These defects appear to be related to reduced EEA1+ early endosomes and accumulation of LC3+ autophagosomes.

I have some remaining concerns, with respect to the new Fig. 6 and Fig. 7.

Fig. 6B & C are inconsistent with each other: Fig.6C, western blot, shows barely any difference in LC3A/B – I & II between control and KO animals at P12 and P14; especially if the slight increase for the KO in the tubulin loading control is accounted for. This is inconsistent with the IF result (panel B), which shows absolutely no LC3A/B signal in the control retina in any retinal cell layers, and yet there is a significant signal in the photoreceptor layer of the KO retina, thus indicating a large control v KO difference.

The IF and immunoEM of p62 is striking. But, especially given the inconsistency in A and B, a western blot should be given for p62.

The increase in both LC3 and p62 is difficult to explain, so that the newly added p62 results may be confusing. However, the authors have been careful in their language, suggesting only "altered autophagy", so the result is not overstated.

The transcriptome analysis figure (Fig.7) does not strongly support the autophagy-related conclusions and perhaps should not be left as a main figure.

Fig. 7B: It appears that the strongest changes are at P6, and in functions related to cell-cycle regulation.

Fig. 7C,D,E: Many of the highlighted genes, while suggesting significant differential expression (pValue < 0.05, -Log10 >1.3), are, nevertheless, not changed much, as their LogFC values are mostly below 0.75 (Log2=0.75), which means there is less than a 2-fold change.

Supplemental Fig. S5B is not mentioned anywhere in the manuscript.

Supplemental Fig. S5C two-colored image on the left does not seem to match the single-colored image on the right – there is clear red signal in at least 2-3 places in the image on the left, yet none is seen in the right image.

Supplemental Fig. S8: if I understood it correctly, the y-axis is showing the –log10 calculation of respective pValue for each of the differentially expressed genes, between Ctrl and KO retinas, highlighted in the graph. If that is the case, most of the genes listed here were not actually differentially expressed between the two genotypes: pValue of 0.05 would yield a –log10 value of around 1.3. Therefore, anything below 1.3 would not be considered significantly differentially expressed. Only 4 or 5 genes in the autophagosome-related pathways were differentially expressed at P6 and perhaps only 1 at P10 and none at P14. I don’t think the gene expression data is very convincing here. The perturbation of autophagy in the KO animal could be real, but it is possible that this is more reflected at the protein level rather than the transcriptome level.

**Have all data underlying the figures and results presented in the manuscript been provided?**

Reviewer #2: Yes

PLOS authors have the option to publish the peer review history of their article (what does this mean?). If published, this will include your full peer review and any attached files.

Reviewer #2: No

---

## [Editor Report · Decision Letter 2]

9 Nov 2020

Dear Anand,

We are pleased to inform you that your manuscript entitled "Loss of endocytosis-associated RabGEF1 causes aberrant morphogenesis and altered autophagy in photoreceptors leading to retinal degeneration" has been editorially accepted for publication in PLOS Genetics. Congratulations!

Yours sincerely,

Uwe 

Guest Editor

PLOS Genetics

Gregory Barsh

Editor-in-Chief

PLOS Genetics

Comments from the reviewers (if applicable):

Uwe Wolfrum (guest editor): The authors have responded to all the criticisms of reviewer 2 regarding the revision of the manuscript and myself, including the criticism of reviewer 3. I also appreciate that the authors followed my suggestion and shortened the title of the manuscript. From my point of view, there is nothing left to prevent publishing the manuscript in PLOSGenetics.

**Data Deposition**

http://datadryad.org/submit?journalID=pgenetics&manu=PGENETICS-D-20-00073R2

**Press Queries**

---

## [Editor Report · Acceptance letter]

3 Dec 2020

PGENETICS-D-20-00073R2 

Loss of endocytosis-associated RabGEF1 causes aberrant morphogenesis and altered autophagy in photoreceptors leading to retinal degeneration 

Dear Dr Swaroop, 

We are pleased to inform you that your manuscript entitled "Loss of endocytosis-associated RabGEF1 causes aberrant morphogenesis and altered autophagy in photoreceptors leading to retinal degeneration" has been formally accepted for publication in PLOS Genetics! Your manuscript is now with our production department and you will be notified of the publication date in due course.

With kind regards,

Nicola Davies

PLOS Genetics

On behalf of:
